# Optimal Best Markovian Arm Identification with Fixed Confidence

**Vrettos Moulos**
University of California Berkeley
vrettos@berkeley.edu

## Abstract

We give a complete characterization of the sampling complexity of best Markovian arm identification in one-parameter Markovian bandit models. We derive instance specific nonasymptotic and asymptotic lower bounds which generalize those of the IID setting. We analyze the Track-and-Stop strategy, initially proposed for the IID setting, and we prove that asymptotically it is at most a factor of four apart from the lower bound. Our one-parameter Markovian bandit model is based on the notion of an exponential family of stochastic matrices for which we establish many useful properties. For the analysis of the Track-and-Stop strategy we derive a novel concentration inequality for Markov chains that may be of interest in its own right.

## 1 Introduction

This paper is about optimal best Markovian arm identification with fixed confidence. There are $K$ independent options which are referred to as arms. Each arm $a$ is associated with a discrete time stochastic process, which is characterized by a parameter $\theta_a$ and it's governed by the probability law $\mathbb{P}_{\theta_a}$. At each round we select one arm, without any prior knowledge of the statistics of the stochastic processes. The stochastic process that corresponds to the selected arm evolves by one time step, and we observe this evolution through a reward function, while the stochastic processes for the rest of the arms stay still. A confidence level $\delta \in (0,1)$ is prescribed, and our goal is to identify the arm that corresponds to the process with the highest stationary mean with probability at least $1 - \delta$, and using as few samples as possible.

### 1.1 Contributions

In the work of Garivier and Kaufmann (2016) the discrete time stochastic process associated with each arm $a$ is assumed to be an IID process. Here we go one step further and we study more complicated dependent processes, which allow us to use more expressive models in the stochastic multi-armed bandits framework. More specifically we consider the case that each $\mathbb{P}_{\theta_a}$ is the law of an irreducible finite state Markov chain associated with a stationary mean $\mu(\theta_a)$. We establish a lower bound (Theorem 1) for the expected sample complexity, as well as an analysis of the Track-and-Stop strategy, proposed for the IID setting in Garivier and Kaufmann (2016), which shows (Theorem 3) that asymptotically the Track-and-Stop strategy in the Markovian dependence setting attains a sample complexity which is at most a factor of four apart from our asymptotic lower bound. Both our lower and upper bounds extend the work of Garivier and Kaufmann (2016) in the more complicated and more general Markovian dependence setting.

The abstract framework of multi-armed bandits has numerous applications in areas like clinical trials, ad placement, adaptive routing, resource allocation, gambling etc. For more context we refer the interested reader to the survey of Bubeck and Cesa-Bianchi (2012). Here we generalize this model to allow for the presence of Markovian dependence, enabling this way the practitioner to use richer and

more expressive models for the various applications. In particular, Markovian dependence allows models where the distribution of next sample depends on the sample just observed. This way one can model for instance the evolution of a rigged slot machine, which as soon as it generates a big reward for the gambler, it changes the reward distribution to a distribution which is skewed towards smaller rewards.

Our key technical contributions stem from the large deviations theory for Markov chains Miller (1961); Donsker and Varadhan (1975); Ellis (1984); Dembo and Zeitouni (1998). In particular we utilize the concept of an *exponential family of stochastic matrices*, first introduced in Miller (1961), in order to model our one-parameter Markovian bandit model. Many properties of the family are established which are then used for our analysis of the Track-and-Stop strategy. The most important one is an optimal concentration inequality for the empirical means of Markov chains (Theorem 2). We are able to establish this inequality for a large class of Markov chains, including those that all the transitions have positive probability. Prior work on the topic, Gillman (1993); Dinwoodie (1995); Lezaud (1998); León and Perron (2004), fails to capture the optimal exponential decay, or introduces a polynomial prefactor, Davisson et al. (1981), as opposed to our constant prefactor. This result may be of independent interest due to the wide applicability of Markov chains in many aspects of learning theory such as various aspects of reinforcement learning, Markov chain Monte Carlo and others.

## 1.2 Related Work

The cornerstone of stochastic multi-armed bandits is the seminal work of Lai and Robbins (1985). They considered $K$ IID process with the objective being to maximize the expected value of the sum of the observed rewards, or equivalently to minimize the so called *regret*. In the same spirit Anantharam et al. (1987a,b) examine the generalization where one is allowed to collect multiple rewards at each time step, first in the case that processes are IID Anantharam et al. (1987a), and then in the case that the processes are irreducible and aperiodic Markov chains Anantharam et al. (1987a). A survey of the regret minimization literature is contained in Bubeck and Cesa-Bianchi (2012).

An alternative objective is the one of identifying the process with the highest stationary mean as fast as and as accurately as possible, notions which are made precise in Subsection 2.1. In the IID setting, Even-Dar et al. (2006) establish an elimination based algorithm in order to find an approximate best arm, and Mannor and Tsitsiklis (0304) provide a matching lower bound. Jamieson et al. (2014) propose an upper confidence strategy, inspired by the law of iterated logarithm, for exact best arm identification given some fixed level of confidence. In the asymptotic high confidence regime, the problem is settled by the work of Garivier and Kaufmann (2016), who provide instance specific matching lower and upper bounds. For their upper bound they propose the Track-and-Stop strategy which is further explored in the work of Kaufmann and Koolen (2018).

The earliest reference for the exponential family of stochastic matrices which is being used to model the Markovian arms can be found in the work of Miller (1961). Exponential families of stochastic matrices lie in the heart of the theory of large deviations for Markov processes, which was popularized with the pioneering work of Donsker and Varadhan (1975). A comprehensive overview of the theory can be found in the book Dembo and Zeitouni (1998). Naturally they also show up when one conditions on the second order empirical distribution of a Markov chain, see the work of Csiszár et al. (1987) about conditional limit theorems. A variant of the exponential family that we are going to discuss has been developed in the context of hypothesis testing in Nakagawa and Kanaya (1993). A more recent development by Nagaoka (2005) gives an information geometry perspective to this concept, and the work Hayashi and Watanabe (2016) examines parameter estimation for the exponential family. Our development of the exponential family of stochastic matrices tries to parallel the development of simple exponential families of probability distributions of Wainwright and Jordan (2008).

Regarding concentration inequalities for Markov chains one of the earliest works Davisson et al. (1981) is based on counting, and is able to capture the optimal rate of exponential decay dictated by the theory of large deviations, but has a suboptimal polynomial prefactor. More recent approaches follow the line of work started by Gillman (1993), who used matrix perturbation theory to derive a bound for reversible Markov chains. This bound attains a constant prefactor but with a suboptimal rate of exponential decay which depends on the spectral gap of the transition matrix. This work was later extended by Dinwoodie (1995); Lezaud (1998) but still with a sub-optimal rate. The work of León and Perron (2004) reduces the problem to a two state Markov chain, and attains the optimal

rate only for the case of a two state Markov chain. Chung et al. (2012) obtain rates that depend on the mixing time of the chain rather than the spectral gap, but which are still suboptimal.

## 2 Problem Formulation

### 2.1 One-parameter family of Markov Chains

In order to model the problem we will use a one-parameter family of Markov chains on a finite state space $S$. Each Markov chain in the family corresponds to a parameter $\theta \in \Theta$, where $\Theta \subseteq \mathbb{R}$ is the parameter space, and is completely characterized by an initial distribution $q_\theta = [q_\theta(x)]_{x \in S}$, and a stochastic transition matrix $P_\theta = [P_\theta(x,y)]_{x,y \in S}$, which satisfy the following conditions.

$$P_\theta \text{ is irreducible for all } \theta \in \Theta. \tag{1}$$
$$P_\theta(x,y) > 0 \Rightarrow P_\lambda(x,y) > 0, \text{ for all } \theta, \lambda \in \Theta, \ x,y \in S. \tag{2}$$
$$q_\theta(x) > 0 \Rightarrow q_\lambda(x), \text{ for all } \theta, \lambda \in \Theta, \ x \in S. \tag{3}$$

There are $K$ Markovian arms with parameters $\boldsymbol{\theta} = (\theta_1, \ldots, \theta_K) \in \Theta^K$, and each arm $a \in [K] = \{1, \ldots, K\}$ evolves as a Markov chain with parameter $\theta_a$ which we denote by $\{X_n^a\}_{x \in \mathbb{Z}_{\geq 0}}$. A non-constant real valued reward function $f : S \to \mathbb{R}$ is applied at each state and produces the reward process $\{Y_n^a\}_{n \in \mathbb{Z}_{\geq 0}}$ given by $Y_n^a = f(X_n^a)$. We can only observe the reward process but not the internal Markov chain. Note that the reward process is a function of the Markov chain and so in general it will have more complicated dependencies than the Markov chain. The reward process is a Markov chain if and only if $f$ is injective. For each $\theta \in \Theta$ there is a unique stationary distribution $\pi_\theta = [\pi_\theta(x)]_{x \in S}$ associated with the stochastic matrix $P_\theta$, due to (1). This allows us to define the stationary reward of the Markov chain corresponding to the parameter $\theta$ as $\mu(\theta) = \sum_x f(x) \pi_\theta(x)$. We will assume that among the $K$ Markovian arms there exists precisely one that possess the highest stationary mean, and we will denote this arm by $a^*(\boldsymbol{\theta})$, so in particular

$$\{a^*(\boldsymbol{\theta})\} = \arg\max_{a \in [K]} \mu(\theta_a).$$

The set of all parameter configurations that possess a unique highest mean is denoted by

$$\boldsymbol{\Theta} = \left\{ \boldsymbol{\theta} \in \Theta^K : \left| \arg\max_{a \in [K]} \mu(\theta_a) \right| = 1 \right\}.$$

The *Kullback-Leibler divergence rate* characterizes the sample complexity of the Markovian identification problem that we are about to study. For two Markov chains of the one-parameter family that are indexed by $\theta$ and $\lambda$ respectively it is given by,

$$D\left(\theta \parallel \lambda\right) = \sum_{x,y \in S} \log \frac{P_\theta(x,y)}{P_\lambda(x,y)} \pi_\theta(x) P_\theta(x,y),$$

where we use the standard notational conventions $\log 0 = \infty$, $\log \frac{\alpha}{0} = \infty$ if $\alpha > 0$, and $0 \log 0 = 0 \ln \frac{0}{0} = 0$. It is always nonnegative, $D\left(\theta \parallel \lambda\right) \geq 0$, with equality occurring if and only if $P_\theta = P_\lambda$, and so $\mu(\theta) \neq \mu(\lambda)$ yields that $D\left(\theta \parallel \lambda\right) > 0$. Furthermore, $D\left(\theta \parallel \lambda\right) < \infty$ due to (2).

With some abuse of notation we will also write $D\left(\mathbb{P} \parallel \mathbb{Q}\right)$ for the Kullback-Leibler divergence between two probability measures $\mathbb{P}$ and $\mathbb{Q}$ on the same measurable space, which is defined as

$$D\left(\mathbb{P} \parallel \mathbb{Q}\right) = \begin{cases} \mathbb{E}_{\mathbb{P}}\left[\log \frac{d\mathbb{P}}{d\mathbb{Q}}\right], & \text{if } \mathbb{P} \ll \mathbb{Q} \\ \infty, & \text{otherwise,} \end{cases}$$

where $\mathbb{P} \ll \mathbb{Q}$ means that $\mathbb{P}$ is absolutely continuous with respect to $\mathbb{Q}$, and in that case $\frac{d\mathbb{P}}{d\mathbb{Q}}$ denotes the Radon-Nikodym derivative of $\mathbb{P}$ with respect to $\mathbb{Q}$.

### 2.2 Best Markovian Arm Identification with Fixed Confidence

Let $\boldsymbol{\theta} \in \boldsymbol{\Theta}$ be an unknown parameter configuration for the $K$ Markovian arms. Let $\delta \in (0,1)$ be a given confidence level. Our goal is to identify $a^*(\boldsymbol{\theta})$ with probability at least $1 - \delta$ using

as few samples as possible. At each time $t$ we select a single arm $A_t$ and we observe the next sample from the reward process $\{Y_n^{A_t}\}_{n \in \mathbb{Z}_{\geq 0}}$, while all the other reward processes stay still. Let $N_a(t) = \sum_{s=1}^t I_{\{A_s=a\}} - 1$ be the number of transitions of the Markovian arm $a$ up to time $t$. Let $\mathcal{F}_t$ be the $\sigma$-field generated by our choices $A_1, \dots, A_t$ and the observations $\{Y_n^1\}_{n=0}^{N_1(t)}, \dots, \{Y_n^a\}_{n=0}^{N_K(t)}$. A sampling strategy, $\mathcal{A}_\delta$, is a triple $\mathcal{A}_\delta = ((A_t)_{t \in \mathbb{Z}_{>0}}, \tau_\delta, \hat{a}_{\tau_\delta})$ consisting of:

- a *sampling rule* $(A_t)_{t \in \mathbb{Z}_{>0}}$, which based on the past decisions and observations $\mathcal{F}_t$, determines which arm $A_{t+1}$ we should sample next, so $A_{t+1}$ is $\mathcal{F}_t$-measurable;

- a *stopping rule* $\tau_\delta$, which denotes the end of the data collection phase and is a stopping time with respect to the filtration $(\mathcal{F}_t)_{t \in \mathbb{Z}_{>0}}$, such that $\mathbb{E}_{\boldsymbol{\lambda}}^{\mathcal{A}_\delta}[\tau_\delta] < \infty$ for all $\boldsymbol{\lambda} \in \boldsymbol{\Theta}$;

- a *decision rule* $\hat{a}_{\tau_\delta}$, which is $\mathcal{F}_{\tau_\delta}$-measurable, and determines the arm that we estimate to be the best one.

Sampling strategies need to perform well across all possible parameter configurations in $\boldsymbol{\Theta}$, therefore we need to restrict our strategies to a class of *uniformly accurate* strategies. This motivates the following standard definition.

*Definition* 1 ($\delta$-PC). Given a confidence level $\delta \in (0,1)$, a sampling strategy $\mathcal{A}_\delta = ((A_t)_{t \in \mathbb{Z}_{>0}}, \tau_\delta, \hat{a}_{\tau_\delta})$ is called $\delta$-PC (Probably Correct) if,

$$\mathbb{P}_{\boldsymbol{\lambda}}^{\mathcal{A}_\delta}(\hat{a}_{\tau_\delta} \neq a^*(\boldsymbol{\lambda})) \leq \delta, \text{ for all } \boldsymbol{\lambda} \in \boldsymbol{\Theta}.$$

Therefore our goal is to study the quantity,

$$\inf_{\mathcal{A}_\delta : \delta - PC} \mathbb{E}_{\boldsymbol{\theta}}^{\mathcal{A}_\delta}[\tau_\delta],$$

both in terms of finding a lower bound, i.e. establishing that no $\delta$-PC strategy can have expected sample complexity less than our lower bound, and also in terms of finding an upper bound, i.e. a $\delta$-PC strategy with very small expected sample complexity. We will do so in the high confidence regime of $\delta \to 0$, by establishing instance specific lower and upper bounds which differ just by a factor of four.

## 3 Lower Bound on the Sample Complexity

Deriving lower bounds in the multi-armed bandits setting is a task performed by change of measure arguments initial introduced by Lai and Robbins (1985). Those change of measure arguments capture the simple idea that in order to identify the best arm we should at least be able to differentiate between two bandit models that exhibit different best arms but are statistically similar. Fix $\theta \in \boldsymbol{\Theta}$, and define the set of parameter configurations that exhibit as best arm an arm different than $a^*(\boldsymbol{\theta})$ by

$$\text{Alt}(\boldsymbol{\theta}) = \{\boldsymbol{\lambda} \in \boldsymbol{\Theta} : a^*(\boldsymbol{\lambda}) \neq a^*(\boldsymbol{\theta})\}.$$

Then we consider an alternative parametrization $\boldsymbol{\lambda} \in \text{Alt}(\boldsymbol{\theta})$ and we write their log-likelihood ratio up to time $t$

$$\log\left(\frac{d\,\mathbb{P}_{\boldsymbol{\theta}}^{\mathcal{A}_\delta} \mid \mathcal{F}_t}{d\,\mathbb{P}_{\boldsymbol{\lambda}}^{\mathcal{A}_\delta} \mid \mathcal{F}_t}\right) = \sum_{a=1}^K I_{\{N_a(t) \geq 0\}} \log \frac{q_{\theta_a}(X_0^a)}{q_{\lambda_a}(X_0^a)} \tag{4}$$
$$+ \sum_{a=1}^K \sum_{x,y} N_a(x,y,0,t) \log \frac{P_{\theta_a}(x,y)}{P_{\lambda_a}(x,y)},$$

where $N_a(x,y,0,t) = \sum_{s=0}^{t-1} 1\{X_s^a = x, X_{s+1}^a = y\}$. The log-likelihood ratio enables us to perform changes of measure for fixed times $t$, and more generally for stopping times $\tau$ with respect to $(\mathcal{F}_t)_{t \in \mathbb{Z}_{>0}}$, which are $\mathbb{P}_{\boldsymbol{\theta}}^{\mathcal{A}_\delta}$ and $\mathbb{P}_{\boldsymbol{\lambda}}^{\mathcal{A}_\delta}$-a.s. finite, through the following change of measure formula,

$$\mathbb{P}_{\boldsymbol{\lambda}}^{\mathcal{A}_\delta}(\mathcal{E}) = \mathbb{E}_{\boldsymbol{\theta}}^{\mathcal{A}_\delta}\left[I_{\mathcal{E}} \frac{d\,\mathbb{P}_{\boldsymbol{\lambda}} \mid \mathcal{F}_\tau}{d\,\mathbb{P}_{\boldsymbol{\theta}} \mid \mathcal{F}_\tau}\right], \text{ for any } \mathcal{E} \in \mathcal{F}_\tau. \tag{5}$$

In order to derive our lower bound we use a technique developed for the IID case by Garivier and Kaufmann (2016) which combines several changes of measure at once. To make this technique work in the Markovian setting we need the following inequality which we derive in Appendix A using a renewal argument for Markov chains.

**Lemma 1.** *Let $\boldsymbol{\theta} \in \boldsymbol{\Theta}$ and $\boldsymbol{\lambda} \in Alt(\boldsymbol{\theta})$ be two parameter configurations. Let $\tau$ be a stopping time with respect to $(\mathcal{F}_t)_{t \in \mathbb{Z}_{>0}}$, with $\mathbb{E}_{\boldsymbol{\theta}}^{\mathcal{A}_\delta}[\tau]$, $\mathbb{E}_{\boldsymbol{\lambda}}^{\mathcal{A}_\delta}[\tau] < \infty$. Then*

$$D\left(\mathbb{P}_{\boldsymbol{\theta}}^{\mathcal{A}_\delta} \mid_{\mathcal{F}_\tau} \middle\| \mathbb{P}_{\boldsymbol{\lambda}}^{\mathcal{A}_\delta} \mid_{\mathcal{F}_\tau}\right)$$

$$\leq \sum_{a=1}^K D\left(q_{\theta_a} \| q_{\lambda_a}\right) + \sum_{a=1}^K \left(\mathbb{E}_{\boldsymbol{\theta}}^{\mathcal{A}_\delta}[N_a(\tau)] + R_{\theta_a}\right) D\left(\theta_a \| \lambda_a\right),$$

*where $R_{\theta_a} = \mathbb{E}_{\theta_a}\left[\inf\{n > 0 : X_n^a = X_0^a\}\right] < \infty$, the first summand is finite due to (3), and the second summand is finite due to (2).*

Combining those ingredients with the data processing inequality we derive our instance specific lower bound for the Markovian bandit identification problem in Appendix A.

**Theorem 1.** *Assume that the one-parameter family of Markov chains on the finite state space $S$ satisfies conditions (1), (2), and (3). Fix $\delta \in (0,1)$, let $f : S \to \mathbb{R}$ be a nonconstant reward function, let $\mathcal{A}_\delta$ be a $\delta$-PC sampling strategy, and fix a parameter configuration $\boldsymbol{\theta} \in \boldsymbol{\Theta}$. Then*

$$T^*(\boldsymbol{\theta}) \leq \liminf_{\delta \to 0} \frac{\mathbb{E}_{\boldsymbol{\theta}}^{\mathcal{A}_\delta}[\tau_\delta]}{\log \frac{1}{\delta}},$$

*where*

$$T^*(\boldsymbol{\theta})^{-1} = \sup_{\boldsymbol{w} \in \mathcal{M}_1([K])} \inf_{\boldsymbol{\lambda} \in Alt(\boldsymbol{\theta})} \sum_{a=1}^K w_a D\left(\theta_a \| \lambda_a\right),$$

*and $\mathcal{M}_1([K])$ denotes the set of all probability distributions on $[K]$.*

As noted in Garivier and Kaufmann (2016) the sup in the definition of $T^*(\boldsymbol{\theta})$ is actually attained uniquely, and therefore we can define $\boldsymbol{w}^*(\boldsymbol{\theta})$ as the unique maximizer,

$$\{\boldsymbol{w}^*(\boldsymbol{\theta})\} = \underset{\boldsymbol{w} \in \mathcal{M}_1([K])}{\arg\max} \inf_{\boldsymbol{\lambda} \in \text{Alt}(\boldsymbol{\theta})} \sum_{a=1}^K w_a D\left(\theta_a \| \lambda_a\right).$$

## 4 One-Parameter Exponential Family of Markov Chains

### 4.1 Definition and Basic Properties

In this section we instantiate the abstract one-parameter family of Markov chains from Subsection 2.1, with the one-parameter exponential family of Markov chains. Given the finite state space $S$, and the nonconstant reward function $f : S \to \mathbb{R}$, we define $M = \max_x f(x)$ and $m = \min_x f(x)$. Based on $f$ we construct two subsets of state space, $S_M = \{x \in S : f(x) = M\}$ and $S_m = \{x \in S : f(x) = m\}$, corresponding to states of maximum and minimum $f$-value respectively. Our goal is to create a family of Markov chains which can realize any stationary mean in the interval $(m, M)$, which will be later used in order to model the Markovian arms. Towards this goal we use as a *generator* for our family, an irreducible stochastic matrix $P$ which satisfies the following properties.

> The submatrix of $P$ with rows and columns in $S_M$ is irreducible. (6)
>
> For every $x \in S - S_M$, there is a $y \in S_M$ such that $P(x,y) > 0$. (7)
>
> The submatrix of $P$ with rows and columns in $S_m$ is irreducible. (8)
>
> For every $x \in S - S_m$, there is a $y \in S_m$ such that $P(x,y) > 0$. (9)

For example, a positive stochastic matrix, i.e. one where all the transition probabilities are positive, satisfies all those properties. Note that in practice this can always be attained by substituting zero transition probabilities with $\epsilon$ transition probabilities, where $\epsilon \in (0,1)$ is some small constant.

Our parameter space will be the whole real line, $\Theta = \mathbb{R}$. Given a parameter $\theta \in \Theta$, we pick an arbitrary initial distribution $q_\theta \in \mathcal{M}_1(S)$ such that $q_\theta(x) > 0$ for all $x \in S$, and we tilt exponentially all the the transitions of $P$ by constructing the matrix $\tilde{P}_\theta(x,y) = P(x,y)e^{\theta f(y)}$. Note that $\tilde{P}_\theta$ is not a stochastic matrix, but we can normalize it and turn it into a stochastic matrix by invoking the Perron-Frobenius theory. Let $\rho(\theta)$ be the spectral radius of $\tilde{P}_\theta$. From the Perron-Frobenius theory

we know that $\rho(\theta)$ is a simple eigenvalue of $\tilde{P}_\theta$, called the Perron-Frobenius eigenvalue, associated with unique left and right eigenvectors $u_\theta$, $v_\theta$ such that they are both positive, $\sum_x u_\theta(x) = 1$, and $\sum_x u_\theta(x)v_\theta(x) = 1$, see for instance Theorem 8.4.4 in the book Horn and Johnson (2013). Let $A(\theta) = \log\rho(\theta)$ be the log-Perron-Frobenius eigenvalue, a quantity which plays a role similar to that of a log-moment-generating function. From $\tilde{P}_\theta$ we can construct an irreducible nonnegative matrix

$$P_\theta(x,y) = \frac{\tilde{P}_\theta(x,y)v_\theta(y)}{\rho(\theta)v_\theta(x)} = \frac{v_\theta(y)}{v_\theta(x)}e^{\theta\phi(y)-A(\theta)}P(x,y),$$

which is stochastic, since

$$\sum_y P_\theta(x,y) = \frac{1}{\rho(\theta)v_\theta(x)} \cdot \sum_y \tilde{P}_\theta(x,y)v_\theta(y) = 1.$$

In addition its stationary distributions is given by

$$\pi_\theta(x) = u_\theta(x)v_\theta(x),$$

since

$$\sum_x \pi_\theta(x)P_\theta(x,y) = \frac{v_\theta(y)}{\rho(\theta)} \cdot \sum_x u_\theta(x)\tilde{P}_\theta(x,y) = u_\theta(y)v_\theta(y) = \pi_\theta(y).$$

Note that the generator stochastic matrix $P$, is the member of the family that corresponds to $\theta = 0$, i.e. $P = P_0, \rho(0) = 1$, and $A(0) = 0$.

The following lemma, whose proof is presented in Appendix B, suggests that the family can be reparametrized using the mean parameters $\mu(\theta)$. More specifically $\dot{A}$ is a strictly increasing bijection between the set $\Theta$ of canonical parameters and the set $\mathcal{M} = \{\mu \in (m, M) : \mu(\theta) = \mu, \text{ for some } \theta \in \Theta\}$ of mean parameters. Therefore with some abuse of notation, we will write $u_\mu, v_\mu, P_\mu, \pi_\mu$ for $u_{\dot{A}^{-1}(\mu)}, v_{\dot{A}^{-1}(\mu)}, P_{\dot{A}^{-1}(\mu)}, \pi_{\dot{A}^{-1}(\mu)}$, and $D(\mu_1 \parallel \mu_2)$ for $D\left(\dot{A}^{-1}(\mu_1) \parallel \dot{A}^{-1}(\mu_2)\right)$.

**Lemma 2.** *Let $P$ be an irreducible stochastic matrix stochastic matrix on a finite state space $S$ which combined with a real-valued function $f : S \to \mathbb{R}$ satisfies (6), (7), (8) and (9). Then the following properties hold true for the exponential family of stochastic matrices generated by $P$ and $f$.*

    *(a) $\rho(\theta)$, $A(\theta)$, $u_\theta$ and $v_\theta$ are analytic functions of $\theta$ on $\Theta = \mathbb{R}$.*

    *(b) $\dot{A}(\theta) = \mu(\theta)$, for all $\theta \in \Theta$.*

    *(c) $\dot{A}(\theta)$ is strictly increasing.*

    *(d) $\mathcal{M} = (m, M)$.*

## 4.2 Concentration for Markov Chains

For a Markov chain $\{X_n\}_{n\in\mathbb{Z}_{\geq 0}}$, driven by an irreducible transition matrix $P$ and an initial distribution $q$, the large deviations theory, Miller (1961); Donsker and Varadhan (1975); Ellis (1984); Dembo and Zeitouni (1998), suggests that the probability of the large deviation event $\{f(X_1) + \ldots + f(X_n) \geq n\mu\}$, when $\mu$ is greater or equal than the stationary mean $\mu(0)$, asymptotically is an exponential decay with the rate of the decay given by a Kullback-Leibler divergence rate. In particular Theorem 3.1.2. from Dembo and Zeitouni (1998) in our context can be written as

$$\lim_{n\to\infty} \frac{1}{n}\log\mathbb{P}_0(f(X_1) + \ldots + f(X_n) \geq n\mu) = -A^*(\mu), \text{ for any } \mu \geq \mu(0),$$

where $A^*(\mu) = \sup_{\theta\in\mathbb{R}}\{\theta\mu - A(\theta)\}$ is the convex conjugate of the log-Perron-Frobenius eigenvalue and represents a Kullback-Leibler divergence rate as we illustrate in Lemma 10.

In the following theorem we present a concentration inequality for Markov chains which attains the rate of exponential decay prescribed from the large deviations theory, as well as a constant prefactor which is independent from $\mu$.

**Theorem 2.** *Let $S$ be a finite state space, and let $P$ be an irreducible stochastic matrix on $S$, which combined with a function $f : S \to \mathbb{R}$ satisfies (6), (7), (8), and (9). Fix $\theta \in \mathbb{R}$, and let $\{X_n\}_{n\in\mathbb{Z}_{\geq 0}}$*

*be a Markov chain on $S$, which is driven by $P_\theta$, the stochastic matrix from the exponential family which corresponds to the parameter $\theta$ and has stationary mean $\mu(\theta)$. Then*

$$\mathbb{P}_\theta \left( f(X_1) + \ldots + f(X_n) \geq n\mu \right) \leq C^2 e^{-nD(\mu \parallel \mu(\theta))}, \text{ for } \mu \in [\mu(\theta), M],$$

*where $C = C(P, f)$ is a constant depending only on the generator stochastic matrix $P$ and the function $f$. In particular, if $P$ is a positive stochastic matrix then we can take $C = \max_{x,y,z} \frac{P(y,z)}{P(x,z)}$.*

We note that in the special case that the process is an IID process the constant $C(P, \phi)$ can be taken to be 1, and thus Theorem 2 generalizes the classic Cramer-Chernoff bound, Chernoff (1952). Observe also that Theorem 2 has a straightforward counterpart for the lower tail as well.

Moreover our inequality is optimal up to the constant prefactor, since the exponential decay is unimprovable due to the large deviations theory, while with respect to the prefactor we can not expect anything better than a constant because otherwise we would contradict the central limit theorem for Markov chains. In particular, when our conditions on $P$ and $f$ are met, our bound dominates similar bounds given by Davisson et al. (1981); Gillman (1993); Dinwoodie (1995); Lezaud (1998); León and Perron (2004).

We give a proof of Theorem 2 in Appendix C, where the main techniques involved are a uniform upper bound on the ratio of the entries of the right Perron-Frobenius eigenvector, as well as an approximation of the log-Perron-Frobenius eigenvalue using the log-moment-generating function.

# 5 Upper Bound on the Sample Complexity: the $(\alpha, \delta)$-Track-and-Stop Strategy

The $(\alpha, \delta)$-Track-and-Stop strategy, which was proposed in Garivier and Kaufmann (2016) in order to tackle the IID setting, tries to track the optimal weights $w_a^*(\boldsymbol{\theta})$. In the sequel we will also write $\boldsymbol{w}^*(\boldsymbol{\mu})$, with $\boldsymbol{\mu} = (\mu(\theta_1), \ldots, \mu(\theta_K))$, to denote $\boldsymbol{w}^*(\boldsymbol{\theta})$. Not having access to $\boldsymbol{\mu}$, the $(\alpha, \delta)$-Track-and-Stop strategy tries to approximate $\boldsymbol{\mu}$ using sample means. Let $\hat{\boldsymbol{\mu}}(t) = (\hat{\mu}_1(N_1(t)), \ldots, \hat{\mu}_K(N_K(t)))$ be the sample means of the $K$ Markov chains when $t$ samples have been observed overall and the calculation of the very first sample from each Markov chain is excluded from the calculation of its sample mean, i.e.

$$\hat{\mu}_a(t) = \frac{1}{N_a(t)} \sum_{s=1}^{N_a(t)} Y_s^a.$$

By imposing sufficient *exploration* the law of large numbers for Markov chains will kick in and the sample means $\hat{\boldsymbol{\mu}}(t)$ will almost surely converge to the true means $\boldsymbol{\mu}$, as $t \to \infty$.

We proceed by briefly describing the three components of the $(\alpha, \delta)$-Track-and-Stop strategy.

## 5.1 Sampling Rule: Tracking the Optimal Proportions

For initialization reasons the first $2K$ samples that we are going to observe are $Y_0^1, Y_1^1, \ldots, Y_0^K, Y_1^K$. After that, for $t \geq 2K$ we let $U_t = \{a : N_a(t) < \sqrt{t} - K/2\}$ and we follow the tracking rule:

$$A_{t+1} \in \begin{cases} \underset{a \in U_t}{\arg\min} \ N_a(t), & \text{if } U_t \neq \emptyset \quad \text{(forced exploration)}, \\ \underset{a=1,\ldots,K}{\arg\max} \ \left\{ w_a^*(\hat{\boldsymbol{\mu}}(t)) - \frac{N_a(t)}{t} \right\}, & \text{otherwise} \quad \text{(direct tracking)}. \end{cases}$$

The forced exploration step is there to ensure that $\hat{\boldsymbol{\mu}}(t) \overset{a.s.}{\to} \boldsymbol{\mu}$ as $t \to \infty$. Then the continuity of $\boldsymbol{\mu} \mapsto \boldsymbol{w}^*(\boldsymbol{\mu})$, combined with the direct tracking step guarantees that almost surely the frequencies $\frac{N_a(t)}{t}$ converge to the optimal weights $w_a^*(\boldsymbol{\mu})$ for all $a = 1, \ldots, K$.

## 5.2 Stopping Rule: $(\alpha, \delta)$-Chernoff's Stopping Rule

For the stopping rule we will need the following statistics. For any two distinct arms $a, b$ if $\hat{\mu}_a(N_a(t)) \geq \hat{\mu}_b(N_b(t))$, we define

$$Z_{a,b}(t) = \frac{N_a(t)}{N_a(t) + N_b(t)} D\left(\hat{\mu}_a(N_a(t)) \parallel \hat{\mu}_{a,b}(N_a(t), N_b(t))\right) +$$

$$\frac{N_b(t)}{N_a(t) + N_b(t)} D\left(\hat{\mu}_b(N_b(t)) \parallel \hat{\mu}_{a,b}(N_a(t), N_b(t))\right),$$

while if $\hat{\mu}_a(N_a(t)) < \hat{\mu}_b(N_b(t))$, we define $Z_{a,b}(t) = -Z_{b,a}(t)$, where

$$\hat{\mu}_{a,b}(N_a(t), N_b(t)) = \frac{N_a(t)}{N_a(t) + N_b(t)} \hat{\mu}_a(N_a(t)) + \frac{N_b(t)}{N_a(t) + N_b(t)} \hat{\mu}_b(N_b(t)).$$

Note that the statistics $Z_{a,b}(t)$ do not arise as the closed form solutions of the Generalized Likelihood Ratio statistics for Markov chains, as it is the case in the IID bandits setting.

For a confidence level $\delta \in (0, 1)$, and a convergence parameter $\alpha > 1$ we define the $(\alpha, \delta)$-Chernoff stopping rule following Garivier and Kaufmann (2016)

$$\tau_{\alpha, \delta} = \inf \left\{ t \in \mathbb{Z}_{>0} : \exists a \in \{1, \ldots, K\} \; \forall b \neq a, \; Z_{a,b}(t) > (0 \vee \beta_{\alpha, \delta}(t)) \right\},$$

where $\beta_{\alpha, \delta}(t) = 2 \log \frac{Dt^\alpha}{\delta}$, $D = \frac{2\alpha K C^2}{\alpha - 1}$, and $C = C(P, f)$ is the constant from Lemma 11. In the special case that $P$ is a positive stochastic matrix we can explicitly set $C = \max_{x,y,z} \frac{P(y,z)}{P(x,z)}$. It is important to notice that the constant $C = C(P, f)$ does not depend on the bandit instance $\boldsymbol{\theta}$ or the confidence level $\delta$, but only on the generator stochastic matrix $P$ and the reward function $f$. In other words it is a characteristic of the exponential family of Markov chains and not of the particular bandit instance, $\boldsymbol{\theta}$, under consideration.

## 5.3 Decision Rule: Best Sample Mean

For a fixed arm $a$ it is clear that, $\min_{b \neq a} Z_{a,b}(t) > 0$ if and only if $\hat{\mu}_a(N_a(t)) > \hat{\mu}_b(N_b(t))$ for all $b \neq a$. Hence the following simple decision rule is well defined when used in conjunction with the $(\alpha, \delta)$-Chernoff stopping rule:

$$\{\hat{a}_{\tau_{\alpha,\delta}}\} = \underset{a=1,\ldots,K}{\arg \max} \; \hat{\mu}_a(N_a(\tau_{\tau_{\alpha,\delta}})).$$

## 5.4 Sample Complexity Analysis

In this section we establish that the $(\alpha, \delta)$-Track-and-Stop strategy is $\delta$-PC, and we upper bound its expected sample complexity. In order to do this we use our Markovian concentration bound Theorem 2.

We first use it in order to establish the following uniform deviation bound.

**Lemma 3.** *Let $\boldsymbol{\theta} \in \boldsymbol{\Theta}$, $\delta \in (0, 1)$, and $\alpha > 1$. Let $\mathcal{A}_\delta$ be a sampling strategy that uses an arbitrary sampling rule, the $(\alpha, \delta)$-Chernoff's stopping rule and the best sample mean decision rule. Then, for any arm $a$,*

$$\mathbb{P}_{\boldsymbol{\theta}}^{\mathcal{A}_\delta}\left(\exists t \in \mathbb{Z}_{>0} : N_a(t) D\left(\hat{\mu}_a(N_a(t)) \parallel \mu_a\right) \geq \beta_{\alpha,\delta}(t)/2\right) \leq \frac{\delta}{K}.$$

With this in our possession we are able to prove in Appendix D that the $(\alpha, \delta)$-Track-and-Stop strategy is $\delta$-PC.

**Proposition 1.** *Let $\delta \in (0, 1)$, and $\alpha \in (1, e/4]$. The $(\alpha, \delta)$-Track-and-Stop strategy is $\delta$-PC.*

Finally, we obtain that in the high confidence regime, $\delta \to 0$, the $(\alpha, \delta)$-Track-and-Stop strategy has a sample complexity which is at most $4\alpha$ times the asymptotic lower bound that we established in Theorem 1.

**Theorem 3.** *Let $\boldsymbol{\theta} \in \boldsymbol{\Theta}$, and $\alpha \in (1, e/4]$. The $(\alpha, \delta)$-Track-and-Stop strategy, denoted here by $\mathcal{A}_\delta$, has its asymptotic expected sample complexity upper bounded by,*

$$\limsup_{\delta \to 0} \frac{\mathbb{E}_{\boldsymbol{\theta}}^{\mathcal{A}_\delta}[\tau_{\alpha,\delta}]}{\log \frac{1}{\delta}} \leq 4\alpha T^*(\boldsymbol{\theta}).$$

## Acknowledgements

We would like to thank Venkat Anantharam, Jim Pitman and Satish Rao for many helpful discussions. This research was supported in part by the NSF grant CCF-1816861.

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
