[Supplementary Material]

# Appendix A Lower Bound on the Sample Complexity

We first prove Lemma 1, for which we will apply a renewal argument. Using the *strong Markov property* we can derive the following standard, see Durrett (2010), decomposition of a Markov chain in IID blocks.

*Fact* 1. Let $\{X_n\}_{n \in \mathbb{Z}_{\geq 0}}$ be an irreducible Markov chain with initial distribution $q$, and transition matrix $P$. Define recursively the $k$-th return time to the initial state as

$$
\begin{cases}
\tau_0 &= 0 \\
\tau_k &= \inf \{n > \tau_{k-1} : X_n = X_0\}, \text{ for } k \geq 1,
\end{cases}
$$

and for $k \geq 1$ let $r_k = \tau_k - \tau_{k-1}$ be the residual time. Those random times partition the Markov chain in a sequence $\{v_k\}_{k \in \mathbb{Z}_{>0}}$ of IID random blocks given by

$$
v_k = (r_k, X_{\tau_{k-1}}, \ldots, X_{\tau_k - 1}), \text{ for } k \geq 1.
$$

Let $N(x, n, m)$ be the number of visits to $x$ that occurred from time $n$ up to time $m$, and $N(x, y, n, m)$ to be the number of transitions from $x$ to $y$ that occurred from time $n$ up to time $m$

$$
N(x, n, m) = \sum_{s=n}^{m-1} 1\{X_s = x\},
$$

$$
N(x, y, n, m) = \sum_{s=n}^{m-1} 1\{X_s = x, X_{s+1} = y\}.
$$

It is well know, see Durrett (2010), that the stationary distribution $\pi$ of the Markov chain is given by

$$
\pi(x) = \frac{\mathbb{E}_{(q,P)} \, N(x, 0, \tau_1)}{\mathbb{E}_{(q,P)} \, \tau_1}, \text{ for any } x \in S. \tag{10}
$$

In the following lemma we establish a similar relation for the invariant distribution over pairs of the Markov chain.

**Lemma 4.**

$$
\pi(x)P(x, y) = \frac{\mathbb{E}_{(q,P)} \, N(x, y, 0, \tau_1)}{\mathbb{E}_{(q,P)} \, \tau_1}, \text{ for any } x, y \in S.
$$

*Proof.* Using (10) it is enough to show that for any initial state $x_0$,

$$
\mathbb{E}_{(x_0,P)} \, N(x, 0, \tau_1)P(x, y) = \mathbb{E}_{(x_0,P)} \, N(x, y, 0, \tau_1),
$$

or equivalently that,

$$
\mathbb{E}_{(x_0,P)} \sum_{n=0}^{\tau_1 - 1} 1\{X_n = x\}P(x, y) = \mathbb{E}_{(x_0,P)} \sum_{n=0}^{\tau_1 - 1} 1\{X_n = x, X_{n+1} = y\}.
$$

Conditioning over the possible values of $\tau_1$, and using Fubini's Theorem we obtain

$$
\mathbb{E}_{(x_0,P)} \sum_{n=0}^{\tau_1 - 1} 1\{X_n = x\}P(x, y) = \sum_{t=1}^{\infty} \mathbb{P}_{x_0}(\tau_1 = t) \sum_{n=0}^{t-1} \mathbb{P}_{(x_0,P)}(X_n = x \mid \tau_1 = t)P(x, y)
$$

$$
= \sum_{n=0}^{\infty} \sum_{t=n+1}^{\infty} \mathbb{P}_{(x_0,P)}(X_n = x, \tau_1 = t)P(x, y)
$$

$$
= \sum_{n=0}^{\infty} \mathbb{P}_{(x_0,P)}(X_n = x, \tau_1 > n)P(x, y)
$$

$$
= \sum_{n=0}^{\infty} \mathbb{P}_{(x_0,P)}(X_n = x, X_{n+1} = y) \, \mathbb{P}_{(x_0,P)}(\tau_1 > n \mid X_n = x)
$$

$$
= \sum_{n=0}^{\infty} \mathbb{P}_{(x_0,P)}(X_n = x, X_{n+1} = y, \tau_1 > n)
$$

$$
= \mathbb{E}_{(x_0,P)} \sum_{n=0}^{\tau_1 - 1} 1\{X_n = x, X_{n+1} = y\},
$$

where the second to last equality holds true due to the reversed Markov property

$$\mathbb{P}_{(x_0,P)}(\tau_1 > n \mid X_n = x, X_{n+1} = y) = \mathbb{P}_{(x_0,P)}(\tau_1 > n \mid X_n = x).$$

$\square$

The following Lemma, which is a variant of Lemma 2.1 in Anantharam et al. (1987b), is the place where we use the IID block structure of the Markov chain.

**Lemma 5.** *Define the mean return time of the Markov chain with initial distribution q and irreducible transition matrix P by*

$$R = \mathbb{E}_{(q,P)}[\inf \{n > 0 : X_n = X_0\}] < \infty.$$

*Let $\mathcal{F}_n$ be the $\sigma$-field generated by $X_0, X_1, \ldots, X_n$. Let $\tau$ be a stopping time with respect to $(\mathcal{F}_n)_{n \in \mathbb{Z}_{\geq 0}}$, with $\mathbb{E}_{(q,P)} \tau < \infty$. Then*

$$\mathbb{E}_{(q,P)} N(x,y,0,\tau) \leq \pi(x)P(x,y)(\mathbb{E}_{(q,P)} \tau + R - 1), \text{ for all } x,y \in S.$$

*Proof.* Using the $k$-th return times from Fact 1 we decompose $N(x,y,0,\tau_k)$ in $k$ IID summands

$$N(x,y,0,\tau_k) = \sum_{i=0}^{k-1} N(x,y,\tau_i,\tau_{i+1}).$$

Now let $\kappa = \inf \{k > 0 : \tau_k \geq \tau\}$, so that $\tau_\kappa$ is the first return time to the initial state after or at time $\tau$. By definition of $\tau_\kappa$ we have that

$$\tau_\kappa - \tau \leq \tau_\kappa - \tau_{\kappa-1} - 1.$$

Taking expectations we obtain

$$\mathbb{E}_{(q,P)}[\tau_\kappa - \tau] \leq \mathbb{E}_{(q,P)}[\tau_\kappa - \tau_{\kappa-1}] - 1 = \mathbb{E}_{(q,P)} r_\kappa - 1 = \mathbb{E}_{(q,P)} r_1 - 1 = R - 1,$$

which also gives that

$$\mathbb{E}_{(q,P)}[\tau_\kappa] \leq \mathbb{E}_{(q,P)}[\tau] + R - 1 < \infty.$$

This allows us to use Wald's identity, followed by Lemma 4, followed by Wald's identity again, in order to get

$$\begin{aligned}
\mathbb{E}_{(q,P)} N(x,y,0,\tau_\kappa) &= \mathbb{E}_{(q,P)} \sum_{i=0}^{\kappa-1} N(x,y,\tau_i,\tau_{i+1}) \\
&= \mathbb{E}_{(q,P)}[N(x,y,0,\tau_1)] \, \mathbb{E}_q[\kappa] \\
&= p(x)P(x,y) \, \mathbb{E}_{(q,P)}[\tau_1] \, \mathbb{E}_{(q,P)}[\kappa] \\
&= p(x)P(x,y) \, \mathbb{E}_{(q,P)}[\tau_\kappa].
\end{aligned}$$

Therefore,

$$\begin{aligned}
\mathbb{E}_{(q,P)} N(x,y,0,\tau) &\leq \mathbb{E}_{(q,P)} N(x,y,0,\tau_\kappa) \\
&= \pi(x)P(x,y) \, \mathbb{E}_{(q,P)}[\tau_\kappa] \\
&\leq \pi(x)P(x,y)(\mathbb{E}_{(q,P)}[\tau] + R - 1).
\end{aligned}$$

$\square$

*Proof of Lemma 1.*

Follows by taking $\mathbb{E}_{\boldsymbol{\theta}}^{\mathcal{A}_\delta}$ of the log-likelihood ratio, $\log \left( \frac{\mathbb{P}_{\boldsymbol{\theta}}^{\mathcal{A}_\delta}|_{\mathcal{F}_\tau}}{\mathbb{P}_{\boldsymbol{\lambda}}^{\mathcal{A}_\delta}|_{\mathcal{F}_\tau}} \right)$, given by (4), and applying Lemma 5 $K$ times for the stopping times $N_a(\tau) + 1, \; a = 1, \ldots, K$. $\square$

The last part of Appendix A involves the proof of Theorem 1.

*Proof of Theorem 1.*
Consider an alternative parametrization $\boldsymbol{\lambda} \in \mathrm{Alt}(\boldsymbol{\theta})$. The data processing inequality, see Cover and Thomas (2006), gives us as a way to lower bound the Kullback-Leibler divergence between the two probability measures $\mathbb{P}_{\boldsymbol{\theta}}^{\mathcal{A}_\delta} |_{\mathcal{F}_{\tau_\delta}}$ and $\mathbb{P}_{\boldsymbol{\lambda}}^{\mathcal{A}_\delta} |_{\mathcal{F}_{\tau_\delta}}$. In particular,

$$D_2 \left( \mathbb{P}_{\boldsymbol{\theta}}^{\mathcal{A}_\delta}(\mathcal{E}) \,\middle\|\, \mathbb{P}_{\boldsymbol{\lambda}}^{\mathcal{A}_\delta}(\mathcal{E}) \right) \leq D \left( \mathbb{P}_{\boldsymbol{\theta}}^{\mathcal{A}_\delta} |_{\mathcal{F}_{\tau_\delta}} \,\middle\|\, \mathbb{P}_{\boldsymbol{\lambda}}^{\mathcal{A}_\delta} |_{\mathcal{F}_{\tau_\delta}} \right), \text{ for any } \mathcal{E} \in \mathcal{F}_{\tau_\delta},$$

where for $p, q \in [0, 1]$, $D_2 (p \,\|\, q)$ denotes the binary Kullback-Leibler divergence,

$$D_2 (p \,\|\, q) = p \log \frac{p}{q} + (1 - p) \log \frac{1 - p}{1 - q}.$$

We apply this inequality with the event $\mathcal{E} = \{\hat{a}_{\tau_\delta} \neq a^*(\boldsymbol{\theta})\} \in \mathcal{F}_{\tau_\delta}$. The fact that the strategy $\mathcal{A}_\delta$ is $\delta$-PC implies that

$$\mathbb{P}_{\boldsymbol{\theta}}(\mathcal{E}) \leq \delta, \quad \text{and} \quad \mathbb{P}_{\boldsymbol{\lambda}}(\mathcal{E}) \geq 1 - \delta,$$

hence

$$D_2 (\delta \,\|\, 1 - \delta) \leq D \left( \mathbb{P}_{\boldsymbol{\theta}}^{\mathcal{A}_\delta} |_{\mathcal{F}_{\tau_\delta}} \,\middle\|\, \mathbb{P}_{\boldsymbol{\lambda}}^{\mathcal{A}_\delta} |_{\mathcal{F}_{\tau_\delta}} \right).$$

Combining this with Lemma 1 we get that

$$D_2 (\delta \,\|\, 1 - \delta) \leq \sum_{a=1}^K D \left( q_{\theta_a} \,\|\, q_{\lambda_a} \right) + \sum_{a=1}^K \left( \mathbb{E}_{\boldsymbol{\theta}}^{\mathcal{A}_\delta}[N_a(\tau_\delta)] + R_a \right) D \left( \theta_a \,\|\, \lambda_a \right).$$

The fact that $\sum_{a=1}^K N_a(\tau_\delta) \leq \tau_\delta$ gives,

$$D_2 (\delta \,\|\, 1 - \delta) - \sum_{a=1}^K D \left( q_{\theta_a} \,\|\, q_{\lambda_a} \right)$$
$$\leq \left( \mathbb{E}_{\boldsymbol{\theta}}^{\mathcal{A}_\delta}[\tau_\delta] + \sum_{a=1}^K R_a \right) \sum_{a=1}^K \frac{\mathbb{E}_{\boldsymbol{\theta}}^{\mathcal{A}_\delta}[N_a(\tau_\delta)] + R_a}{\sum_{b=1}^K \left( \mathbb{E}_{\boldsymbol{\theta}}^{\mathcal{A}_\delta}[N_b(\tau_\delta)] + R_b \right)} D \left( \theta_a \,\|\, \lambda_a \right),$$

and now we follow the technique of Garivier and Kaufmann (2016) which combines multiple alternative models $\boldsymbol{\lambda}$,

$$D_2 (\delta \,\|\, 1 - \delta) - \sum_{a=1}^K D \left( q_{\theta_a} \,\|\, q_{\lambda_a} \right)$$
$$\leq \left( \mathbb{E}_{\boldsymbol{\theta}}^{\mathcal{A}_\delta}[\tau_\delta] + \sum_{a=1}^K R_a \right) \inf_{\boldsymbol{\lambda} \in \mathrm{Alt}(\boldsymbol{\theta})} \sum_{a=1}^K \frac{\mathbb{E}_{\boldsymbol{\theta}}^{\mathcal{A}_\delta}[N_a(\tau_\delta)] + R_a}{\sum_{b=1}^K \left( \mathbb{E}_{\boldsymbol{\theta}}^{\mathcal{A}_\delta}[N_b(\tau_\delta)] + R_b \right)} D \left( \theta_a \,\|\, \lambda_a \right)$$
$$\leq \left( \mathbb{E}_{\boldsymbol{\theta}}^{\mathcal{A}_\delta}[\tau_\delta] + \sum_{a=1}^K R_a \right) \sup_{\boldsymbol{w} \in \mathcal{M}_1([K])} \inf_{\boldsymbol{\lambda} \in \mathrm{Alt}(\boldsymbol{\theta})} \sum_{a=1}^K w_a D \left( \theta_a \,\|\, \lambda_a \right).$$

The conclusion follows by letting $\delta$ go to 0, and using the fact that

$$\lim_{\delta \to 0} \frac{D_2 (\delta \,\|\, 1 - \delta)}{\log \frac{1}{\delta}} = 1.$$

$\square$

# Appendix B    Exponential Family of Stochastic Matrices

For a stochastic matrix $P$ on $S$, and a probability distribution $p \in \mathcal{M}_1(S)$, we use the notation $p \odot P \in \mathcal{M}_1(S \times S)$ to denote the bivariate distribution on $S \times S$ given by

$$(p \odot P)(x, y) = p(x) P(x, y).$$

We start by establishing parts (a), (b) and (c) of Lemma 2.

*Proof of Lemma 2.*

(a) Each entry of $\tilde{P}_\theta$ is a real analytic function of $\theta$, and for each $\theta_0$ the Perron-Frobenius eigenvalue $\rho(\theta_0)$ is simple with a unique corresponding left and right eigenvectors $u_{\theta_0}$, $v_{\theta_0}$ and such that they are both positive, $\sum_x u_{\theta_0}(x) = 1$ and $\sum_x u_{\theta_0}(x)v_{\theta_0}(x) = 1$. The conclusion follows by standard implicit function theorem type of arguments. See for example Theorem 7 and Theorem 8 in Chapter 9 from the book of Lax (2007).

(b) For any $x, y \in S$ such that $P(x, y) > 0$ we have that

$$\log P_\theta(x, y) = \theta f(y) - A(\theta) + \log v_\theta(y) - \log v_\theta(x) + \log P(x, y).$$

Differentiating with respect to $\theta$, and taking expectation with respect to $\pi_\theta \odot P_\theta$ we obtain

$$\mathbb{E}_{(X,Y)\sim\pi_\theta\odot P_\theta} \frac{d}{d\theta} \log P_\theta(X, Y) = \pi_\theta(f) - \dot{A}(\theta),$$

where the logarithms cancel out since $\pi_\theta \odot P_\theta$ has identical marginals. The conclusion follows because

$$\mathbb{E}_{(X,Y)\sim\pi_\theta\odot P_\theta} \frac{d}{d\theta} \log P_\theta(X, Y) = \sum_x \pi_\theta(x) \frac{d}{d\theta} \left( \sum_y P_\theta(x, y) \right) = 0.$$

(c) For any $x, y \in S$ such that $P(x, y) > 0$ we have that

$$\frac{d^2}{d\theta^2} \log P_\theta(x, y) = -\ddot{A}(\theta) + \frac{d^2}{d\theta^2} \log v_\theta(y) - \frac{d^2}{d\theta^2} \log v_\theta(x).$$

Taking expectation with respect to $\pi_\theta \odot P_\theta$ we obtain

$$\ddot{A}(\theta) = -\mathbb{E}_{(X,Y)\sim\pi_\theta\odot P_\theta} \frac{d^2}{d\theta} \log P_\theta(X, Y)$$

$$= \mathbb{E}_{(X,Y)\sim\pi_\theta\odot P_\theta} \left( \frac{d}{d\theta} \log P_\theta(X, Y) \right)^2 \geq 0.$$

This ensures that $\dot{A}(\theta)$ is increasing.

Assume, towards contradiction, that $\ddot{A}(\theta) = 0$ in a neighborhood of $\theta_0$. Then $P_\theta$ does not depend on $\theta$ in a neighborhood of $\theta_0$. The $S_M$ component is irreducible so we can find $x_1, \ldots, x_{l+1} \in S_M$ such that $P(x_i, x_{i+1}) > 0$ for $i = 1, \ldots, l$ and $x_1 = x_{l+1}$, and so

$$P_\theta(x_1, x_2) \ldots P_\theta(x_l, x_{l+1}) = \frac{P(x_1, x_2) \ldots P(x_l, x_{l+1})e^{\theta l M}}{\rho(\theta)^l},$$

and the $S_m$ component is irreducible as well so we can find $y_1, \ldots, y_{k+1} \in S_m$ such that $P(y_i, y_{i+1}) > 0$ for $i = 1, \ldots, k$ and $y_1 = y_{k+1}$, and so

$$P_\theta(y_1, y_2) \ldots P_\theta(y_l, y_{k+1}) = \frac{P(y_1, y_2) \ldots P(y_k, y_{k+1})e^{\theta k m}}{\rho(\theta)^k}.$$

This means that the ratio

$$\frac{(P_\theta(x_1, x_2) \cdots P_\theta(x_l, x_{l+1}))^{1/l}}{(P_\theta(y_1, y_2) \cdots P_\theta(y_k, y_{k+1}))^{1/k}} = \frac{P(x_1, x_2) \cdots P(x_l, x_{l+1})}{P(y_1, y_2) \cdots P(y_k, y_{k+1})} e^{\theta(M-m)},$$

depends on $\theta$. This contradicts the assumption that $P_\theta$ does not depend on $\theta$ on a neighborhood of $\theta_0$.

Therefore, $\ddot{A}(\theta)$ does not vanish on any nonempty open interval of $\mathbb{R}$, and so we conclude that $\dot{A}(\theta)$ is strictly increasing.

$\square$

Showing part (d) of Lemma 2 requires the study of the limiting behavior of the family which we do in the following two Lemmata. The first is a simple extension of the Perron-Frobenius theory.

**Lemma 6.** *Let $W \in \mathbb{R}_{\geq 0}^{n \times n}$ be a non-negative matrix consisting of: a non-negative irreducible square block $A \in \mathbb{R}_{\geq 0}^{k \times k}$, and a non-negative rectangular block $B \in \mathbb{R}_{\geq 0}^{(n-k) \times k}$ such that none of the rows of $B$ is zero, for some $k \in \{1, \dots, n\}$, assembled together in the following way:*

$$W = \begin{bmatrix} A & 0 \\ B & 0 \end{bmatrix},$$

*Then, $\rho(W) = \rho(A)$ is a simple eigenvalue of $W$, which we call the Perron-Frobenius eigenvalue, and is associated with unique left and right eigenvectors $u_W, v_W$ such that $u_W$ has its first $k$ coordinates positive and its last $n - k$ coordinates equal to zero, $v_W$ is positive, $\sum_{x=1}^{n} u_W(x) = 1$, and $\sum_{x=1}^{n} u_W(x) v_W(x) = 1$.*

*Proof.* Let $u_A, v_A$ be the unique left and right eigenvectors of $A$ corresponding to the Perron-Frobenius eigenvalue $\rho(A)$, such that both of them are positive, $\sum_{x=1}^{k} u_A(x) = 1$ and $\sum_{x=1}^{k} u_A(x) v_A(x) = 1$. Observe that the vectors

$$u_W = \begin{bmatrix} u_A \\ 0 \end{bmatrix}, \text{ and } v_W = \begin{bmatrix} v_A \\ Bv_A/\rho(A) \end{bmatrix},$$

are left and right eigenvectors of $W$ with associated eigenvalue $\rho(A)$, and satisfy all the conditions. In addition, $\rho(W)$ being greater than $\rho(A)$, or $\rho(W)$ not being a simple eigenvalue, or $u_W, v_W$ not being unique would contradict the Perron-Frobenius Theorem for the nonnegative irreducible matrix $A$. $\qquad\square$

Now we define the matrix $\overline{P}_\infty = \lim_{\theta \to \infty} e^{-\theta M} \tilde{P}_\theta$, i.e. the matrix $P$ where we keep the columns $y \in S_M$ intact, and we zero out all the other columns. After suitable permutation of the states Lemma 6 applies for $\overline{P}_\infty$, and so $\rho(\overline{P}_\infty)$ is a simple eigenvalue of $\overline{P}_\infty$, which is associated with unique left and right eigenvectors $u_\infty, v_\infty$ such that $u_\infty(x) > 0$ for $x \in S_M$ and $u_\infty(x) = 0$ for $x \notin S_M$, $v_\infty$ is positive, $\sum_x u_\infty(x) = 1$ and $\sum_x u_\infty(x) v_\infty(x) = 1$. Similarly, we define $\overline{P}_{-\infty} := \lim_{\theta \to -\infty} e^{-\theta m} \tilde{P}_\theta$, with Perron-Frobenius eigenvalue $\rho(\overline{P}_{-\infty})$, which is associated with unique left and right eigenvectors $u_{-\infty}, v_{-\infty}$ such that $u_{-\infty}(x) > 0$ for $x \in S_m$ and $u_{-\infty}(x) = 0$ for $x \notin S_m$, $v_{-\infty}$ is positive, $\sum_x u_{-\infty}(x) = 1$ and $\sum_x u_{-\infty}(x) v_{-\infty}(x) = 1$.

The following Lemma characterizes the limiting stochastic matrices $P_\infty$, $P_{-\infty}$ of the exponential family, and proves part (d) of Lemma 2.

**Lemma 7.**

(a) $\theta M - A(\theta) \to -\log \rho(\overline{P}_\infty)$, $u_\theta \to u_\infty$, $v_\theta \to v_\infty$, *as* $\theta \to \infty$, *and so*

$$\lim_{\theta \to \infty} P_\theta(x, y) = \frac{\overline{P}_\infty(x, y) v_\infty(y)}{\rho(\overline{P}_\infty) v_\infty(x)} =: P_\infty(x, y),$$

*and* $\pi_\theta(f) \to M$ *as* $\theta \to \infty$.

(b) $\theta m - A(\theta) \to -\log \rho(\overline{P}_{-\infty})$, $u_\theta \to u_{-\infty}$, $v_\theta \to v_{-\infty}$, *as* $\theta \to -\infty$, *and so*

$$\lim_{\theta \to -\infty} P_\theta(x, y) = \frac{\overline{P}_{-\infty}(x, y) v_{-\infty}(y)}{\rho(\overline{P}_{-\infty}) v_{-\infty}(x)} =: P_{-\infty}(x, y),$$

*and* $\pi_\theta(f) \to m$ *as* $\theta \to -\infty$.

*Proof.* Both parts are a straightforward application of the continuity of the function $P \mapsto (\rho(P), u_P, v_P)$, at $\overline{P}_\infty$ and $\overline{P}_{-\infty}$. The continuity of eigenvalues and eigenvectors is due to the fact that the Perron-Frobenius eigenvalue $\rho(P)$ is a simple eigenvalue and more details can be found in Chapter 3 of the book Ortega (1990). $\qquad\square$

This lemma suggests that we can extend the domain of $\dot{A}(\theta)$ by continuity over the set of extended real numbers $\overline{\mathbb{R}} = \mathbb{R} \cup \{\pm\infty\}$, by defining $\dot{A}(\infty) = M$ and $\dot{A}(-\infty) = m$. This way we have a one-to-one and onto correspondence of $\overline{\mathbb{R}}$ with the closed interval $[m, M]$, with the limit stochastic

matrices being $P_\infty$ and $P_{-\infty}$, which represent degenerate Markov chains where all the transitions lead into states $y \in S_M$ when $\theta = \infty$, and into states $y \in S_m$ when $\theta = -\infty$.

We proceed by deriving some alternative representations for the Kullback-Leibler divergence rate between elements of the exponential family. The following lemma is needed in order to derive the asymptotic Kullback-Leibler divergence rate.

**Lemma 8.**

(a) $\theta \dot{A}(\theta) - A(\theta) \to -\log \rho(\overline{P}_\infty)$, as $\theta \to \infty$.

(b) $\theta \dot{A}(\theta) - A(\theta) \to -\log \rho(\overline{P}_{-\infty})$, as $\theta \to -\infty$.

*Proof.* Let $M_2 = \max_{x \notin S_M} f(x)$. Fix $x \in S$ and $y \notin S_M$. Pick $y_M \in S_M$ such that $P(x, y_M) > 0$. Using Lemma 11 we see that there is a constant $C = C(P, f)$ such that
$$P_\theta(x, y) \le Ce^{-\theta(M - f(y))} P_\theta(x, y_M) \le Ce^{-\theta(M - M_2)}.$$
Therefore the stationary probability of any such $y$ is at most $\pi_\theta(y) \le Ce^{-\theta(M - M_2)}$, and so
$$\pi_\theta(f) \ge (1 - C|S|e^{-\theta(M - M_2)})M + C|S|e^{-\theta(M - M_2)}m.$$
From this we obtain that
$$0 \le \theta(M - \pi_\theta(f)) \le C|S|\theta e^{-\theta(M - M_2)}(M - m), \text{ for any } \theta \ge 0,$$
which yields that $\theta(\dot{A}(\theta) - M) \to 0$, as $\theta \to \infty$. Part (a) now follows, since Lemma 7 suggests that $\theta M - A(\theta) \to -\log \rho(\overline{P}_\infty)$, as $\theta \to \infty$. The second limit follows by the same argument. $\square$

Having this in our possession we state and prove alternative representations for the Kullback-Leibler divergence rate.

**Lemma 9.**

(a) *For all $\theta_1, \theta_2 \in \mathbb{R}$,*
$$D(\theta_1 \parallel \theta_2) = \theta_1 \dot{A}(\theta_1) - A(\theta_1) - (\theta_2 \dot{A}(\theta_1) - A(\theta_2));$$
$$D(\infty \parallel \theta_2) = -\log \rho(\overline{P}_\infty) - (\theta_2 M - A(\theta_2));$$
$$D(-\infty \parallel \theta_2) = -\log \rho(\overline{P}_{-\infty}) - (\theta_2 m - A(\theta_2)).$$

(b) *For all $\mu_1, \mu_2 \in (m, M)$,*
$$D(\mu_1 \parallel \mu_2) = \dot{A}^{-1}(\mu_1)\mu_1 - A(\dot{A}^{-1}(\mu_1)) - (\dot{A}^{-1}(\mu_2)\mu_1 - A(\dot{A}^{-1}(\mu_2)));$$
$$D(M \parallel \mu_2) = -\log \rho(\overline{P}_\infty) - (\dot{A}^{-1}(\mu_2)M - A(\dot{A}^{-1}(\mu_2)));$$
$$D(m \parallel \mu_2) = -\log \rho(\overline{P}_{-\infty}) - (\dot{A}^{-1}(\mu_2)m - A(\dot{A}^{-1}(\mu_2))).$$

*Proof.* For $\theta_1, \theta_2 \in \mathbb{R}$ we have that
$$D(\theta_1 \parallel \theta_2) = \mathbb{E}_{(X,Y) \sim \pi_{\theta_1} \odot P_{\theta_1}} \log \frac{P_{\theta_1}(X, Y)}{P_{\theta_2}(X, Y)}$$
$$= A(\theta_2) - A(\theta_1) - (\theta_2 - \theta_1)\dot{A}(\theta_1) + \mathbb{E}_{(X,Y) \sim \pi_{\theta_1} \odot P_{\theta_1}} \left[ \log \frac{v_{\theta_1}(Y)}{v_{\theta_1}(X)} - \log \frac{v_{\theta_2}(Y)}{v_{\theta_2}(X)} \right]$$
$$= \theta_1 \dot{A}(\theta_1) - A(\theta_1) - (\theta_2 \dot{A}(\theta_1) - A(\theta_2)),$$
and the third equality follows due to the fact that $\pi_{\theta_1} \odot P_{\theta_1}$ has identical marginals and so the expectation vanishes.

Now let $\theta_2 \in \mathbb{R}$. Using the continuity of the Kullback-Leibler divergence rate, the formula that we just established, and Lemma 8 we obtain
$$D(\infty \parallel \theta) = \lim_{\theta_1 \to \infty} D(\theta_1 \parallel \theta_2)$$
$$= \lim_{\theta_1 \to \infty} \left( \theta_1 \dot{A}(\theta_1) - A(\theta_1) \right) - \lim_{\theta_1 \to \infty} \left( \theta_2 \dot{A}(\theta_1) - A(\theta_2) \right)$$
$$= -\log \rho(\overline{P}_\infty) - (\theta_2 M - A(\theta_2)).$$
We argue in the same way for $D(-\infty \parallel \theta)$, and part (b) directly follows from part (a). $\square$

As a direct consequence of these representation we obtain the following monotonicity properties of the Kullback-Leibler divergence rate.

**Corollary 1.**

(a) *For fixed $\theta_2 \in \mathbb{R}$, the function $\theta_1 \mapsto D\left(\theta_1 \parallel \theta_2\right)$ is strictly increasing in the interval $[\theta_2, \infty]$ and strictly decreasing in the interval $[-\infty, \theta_2]$.*

(b) *For fixed $\mu_2 \in (m, M)$, the function $\mu_1 \mapsto D\left(\mu_1 \parallel \mu_2\right)$ is strictly increasing in the interval $[\mu_2, M]$ and strictly decreasing in the interval $[m, \mu_2]$.*

We close this appendix by establishing that the Kullback-Leibler divergence rate is the convex conjugate of the log-Perron-Frobenius eigenvalue.

**Lemma 10.**

$$D\left(\mu \parallel \mu(0)\right) = \sup_{\theta \in \mathbb{R}} \left\{\theta \mu - A(\theta)\right\} = \begin{cases} \sup_{\theta \geq 0} \left\{\theta \mu - A(\theta)\right\}, & \text{if } \mu \in [\mu(0), M] \\ \sup_{\theta \leq 0} \left\{\theta \mu - A(\theta)\right\}, & \text{if } \mu \in [m, \mu(0)]. \end{cases}$$

*Proof.* Fix $\mu \in (m, M)$. The function $\theta \mapsto \theta \mu - A(\theta)$ is strictly concave and its derivative vanishes at $\theta = \dot{A}^{-1}(\mu)$, which belong in $[0, \infty)$ when $\mu \in [\mu(0), M]$ and in $(-\infty, 0]$ when $\mu \in (m, \mu(0)]$. Therefore, using Lemma 9 we obtain

$$\sup_{\theta \in \mathbb{R}} \left\{\theta \mu - A(\theta)\right\} = \dot{A}^{-1}(\mu)\mu - A(\dot{A}^{-1}(\mu)) = D\left(\mu \parallel \pi(f)\right).$$

Similarly when $\mu = M$ or $\mu = m$, the derivative only vanishes at $\infty$ and $-\infty$ respectively, and so from a combination of Lemma 7 and Lemma 9 we obtain

$$\sup_{\theta \in \mathbb{R}} \left\{\theta M - A(\theta)\right\} = \lim_{\theta \to \infty} \left(\theta M - A(\theta)\right) = D\left(M \parallel \pi(f)\right),$$

and

$$\sup_{\theta \in \mathbb{R}} \left\{\theta m - A(\theta)\right\} = \lim_{\theta \to -\infty} \left(\theta m - A(\theta)\right) = D\left(m \parallel \pi(f)\right).$$

$\square$

## Appendix C   Concentration for Markov Chains

We first use continuity in order to get a uniform bound on the ratio of the entries of the right Perron-Frobenius eigenvector.

**Lemma 11.** *Let $P$ be an irreducible stochastic matrix on $S$, which combined with $f : S \to \mathbb{R}$ satisfies (6), (7), (8), and (9). There exists a constant $C = C(P, \phi) \geq 1$ such that*

$$C^{-1} \leq \sup_{\theta \in \mathbb{R}, x, y \in S} \frac{v_\theta(y)}{v_\theta(x)} \leq C.$$

*If in addition $P$ is a positive stochastic matrix then we can take $C = \max_{x,y,z} \frac{P(y,z)}{P(x,z)}$.*

*Proof.* For any $x, y \in S$, the ratio $\frac{v_\theta(y)}{v_\theta(x)}$ is a positive real number, and due to Lemma 2 a continuous function of $\theta$. In addition Lemma 6 and Lemma 7 suggest that its limit points $\frac{v_\infty(y)}{v_\infty(x)}$, $\frac{v_{-\infty}(y)}{v_{-\infty}(x)}$ are positive real numbers as well, hence we can take $C = \sup_{\theta \in \mathbb{R}, x, y \in S} \frac{v_\theta(y)}{v_\theta(x)} \geq 1$, which is guaranteed to be finite.

In the special case that $P$ is a positive stochastic matrix, we use the fact that $v_\theta$ is a right Perron-Frobenius eigenvector of $\tilde{P}_\theta$ in order to write

$$\frac{v_\theta(y)}{v_\theta(x)} = \frac{\sum_w \tilde{P}_\theta(y, w) v_\theta(w)}{\sum_w \tilde{P}_\theta(x, w) v_\theta(w)}, \text{ for all } x, y \in S.$$

Now using the simple inequality

$$\left( \min_z \frac{\tilde{P}_\theta(y,z)}{\tilde{P}_\theta(x,z)} \right) \tilde{P}_\theta(x,w) \le \tilde{P}_\theta(y,w) \le \left( \max_z \frac{\tilde{P}_\theta(y,z)}{\tilde{P}_\theta(x,z)} \right) \tilde{P}_\theta(x,w), \text{ for all } x,y,w \in S,$$

and observing that $\frac{\tilde{P}_\theta(y,z)}{\tilde{P}_\theta(x,z)} = \frac{P(y,z)}{P(x,z)}$ we obtain

$$\min_z \frac{P(y,z)}{P(x,z)} \le \frac{v_\theta(y)}{v_\theta(x)} \le \max_z \frac{P(y,z)}{P(x,z)}.$$

$\square$

Next we establish a Proposition which gives us an approximation of the log-Perron-Frobenius eigenvalue using the log-moment-generating-function

$$A_n(\theta) = \frac{1}{n} \log \mathbb{E}_0 \exp\left\{ \theta(\phi(X_1) + \ldots + \phi(X_n)) \right\}$$

**Proposition 2.** *Let $P$ be an irreducible stochastic matrix on $S$, which combined with $f : S \to \mathbb{R}$ satisfies* (6)*,* (7)*,* (8)*, and* (9)*. Then*

$$|A_n(\theta) - A(\theta)| \le \frac{\log C}{n}, \text{ for all } \theta \in \mathbb{R},$$

*where $C = C(P,f)$ is the constant from Lemma 11.*

*Proof.* We start with the following calculation

$$e^{nA_n(\theta)} = \sum_{x_0,x_1,\ldots,x_{n-1},x_n} q(x_0) P(x_0,x_1) e^{\theta\phi(x_1)} \cdots P(x_{n-1},x_n) e^{\theta\phi(x_n)}$$
$$= \sum_{x_0,x_n} q(x_0) \tilde{P}_\theta^n(x_0,x_n).$$

From this using the simple inequality

$$\frac{v_\theta(y)}{\max_x v_\theta(x)} \le 1 \le \frac{v_\theta(y)}{\min_x v_\theta(x)}, \text{ for all } y \in S,$$

together with the fact that $v_\theta$ is a right Perron-Frobenius eigenvector of $\tilde{P}_\theta$ we obtain

$$\min_{x,y} \frac{v_\theta(y)}{v_\theta(x)} e^{nA(\theta)} \le e^{nA_n(\theta)} \le \max_{x,y} \frac{v_\theta(y)}{v_\theta(x)} e^{nA(\theta)}.$$

The conclusion now follows by applying Lemma 11 $\square$

One more ingredient that we need is a uniform bound of the constant $C(P_\theta, f)$ over $\theta \in \mathbb{R}$.

**Lemma 12.** *For the constant from Lemma 11 we have that,*

$$\sup_{\theta \in \mathbb{R}} C(P_\theta, f) \le C(P,f)^2.$$

*Proof.* Recall that

$$C(P_{\theta_2}, f) = \sup_{\theta_1 \in \mathbb{R}, x,y \in S} \frac{v_{\widetilde{(P_{\theta_2})}_{\theta_1}}(y)}{v_{\widetilde{(P_{\theta_2})}_{\theta_1}}(x)}.$$

We claim that

$$\frac{v_{\widetilde{(P_{\theta_2})}_{\theta_1}}(y)}{v_{\widetilde{(P_{\theta_2})}_{\theta_1}}(x)} = \frac{v_{\tilde{P}_{\theta_1+\theta_2}}(y) v_{\tilde{P}_{\theta_2}}(x)}{v_{\tilde{P}_{\theta_1+\theta_2}}(x) v_{\tilde{P}_{\theta_2}}(y)}.$$

To see this we just need to verify that

$$v_{\tilde{P}_{\theta_2}}(x)v_{\widetilde{(P_{\theta_2})}_{\theta_1}}(x), \ x \in S,$$

is a right eigenvector of $\tilde{P}_{\theta_1+\theta_2}$, with associated eigenvalue $\rho(\tilde{P}_{\theta_2})\rho\left(\widetilde{(P_{\theta_2})}_{\theta_1}\right)$, which from the Perron-Frobenious theory has to be the Perron-Frobenious eigenvalue since the associated eigenvector has positive entries. The verification is straight forward

$$\sum_y \tilde{P}_{\theta_1+\theta_2}(x,y)v_{\tilde{P}_{\theta_2}}(y)v_{\widetilde{(P_{\theta_2})}_{\theta_1}}(y) = \rho(\tilde{P}_{\theta_2})v_{\tilde{P}_{\theta_2}}(x)\sum_y \widetilde{(P_{\theta_2})}_{\theta_1}(x,y)v_{\widetilde{(P_{\theta_2})}_{\theta_1}}(y)$$

$$= \rho(\tilde{P}_{\theta_2})\rho\left(\widetilde{(P_{\theta_2})}_{\theta_1}\right)v_{\tilde{P}_{\theta_2}}(x)v_{\widetilde{(P_{\theta_2})}_{\theta_1}}(x), \text{ for all } x \in S.$$

From this we see that

$$\sup_{\theta_1,\theta_2\in\mathbb{R},x,y\in S}\frac{v_{\widetilde{(P_{\theta_2})}_{\theta_1}}(y)}{v_{\widetilde{(P_{\theta_2})}_{\theta_1}}(x)} \leq \left(\sup_{\theta_1,\theta_2\in\mathbb{R},x,y\in S}\frac{v_{\tilde{P}_{\theta_1+\theta_2}}(y)}{v_{\tilde{P}_{\theta_1+\theta_2}}(x)}\right)\left(\sup_{\theta_2\in\mathbb{R},x,y\in S}\frac{v_{\tilde{P}_{\theta_2}}(x)}{v_{\tilde{P}_{\theta_2}}(y)}\right) = C(P,f)^2.$$

$\square$

We are now ready to prove [Theorem 2].

*Proof of [Theorem 2].*
We first prove the bound for $\theta = 0$. Fix $\mu \in [\mu(0), M]$, and $\eta \geq 0$.

$$\mathbb{P}_0\left(f(X_1)+\ldots+f(X_n) \geq n\mu\right) \leq \mathbb{P}_0\left(e^{\eta(f(X_1)+\ldots+f(X_n))} \geq e^{\eta n\mu}\right)$$

$$\leq e^{-n(\eta\mu - A_n(\eta))}$$

$$\leq C(P,f)e^{-n(\eta\mu - A(\eta))},$$

where the second inequality is Markov's inequality, and the third is the estimate from [Proposition 2]. By optimizing over $\eta \geq 0$ and applying [Lemma 10], we obtain

$$\mathbb{P}_0\left(f(X_1)+\ldots+f(X_n) \geq n\mu\right) \leq C(P,f)e^{-nD(\mu \parallel \mu(0))}.$$

Applying this bound with $P_\theta$ in place of $P$, and using [Lemma 12] we conclude that for $\mu \in [\mu(\theta), M]$

$$\mathbb{P}_\theta\left(f(X_1)+\ldots+f(X_n) \geq n\mu\right) \leq C(P_\theta,f)e^{-nD(\mu \parallel \mu(\theta))} \leq C(P,f)^2 e^{-nD(\mu \parallel \mu(\theta))}.$$

$\square$

# Appendix D   Upper Bound on the Sample Complexity: the $(\alpha, \delta)$-Track-and-Stop Strategy

The proof of [Lemma 3] uses the concentration bound [Theorem 2], combined with the monotonicity of the Kullback-Leibler divergence rate [Corollary 1].

*Proof of [Lemma 3].*
We first note the following inclusion of events

$$\bigcup_{t=1}^\infty \bigcup_{n=1}^t \left\{N_a(t)D\left(\hat{\mu}_a(N_a(t)) \parallel \mu_a\right) \geq \beta_{\alpha,\delta}(t)/2, \ N_a(t) = n\right\}$$

$$\subseteq \bigcup_{t=1}^\infty \bigcup_{n=1}^t \left\{nD\left(\hat{\mu}_a(n) \parallel \mu_a\right) \geq \beta_{\alpha,\delta}(t)/2\right\}$$

$$= \bigcup_{t=1}^\infty \left\{tD\left(\hat{\mu}_a(t) \parallel \mu_a\right) \geq \beta_{\alpha,\delta}(t)/2\right\},$$

where the last equality follows because, by the monotonicity of $t \mapsto \beta_{\alpha,\delta}(t)/2$ we have that for each $n \in \mathbb{Z}_{>0}$ and for each $t = n, n+1, \ldots$

$$\{nD\left(\hat{\mu}_a(n) \parallel \mu_a\right) \geq \beta_{\alpha,\delta}(t)/2\} \subseteq \{nD\left(\hat{\mu}_a(n) \parallel \mu_a\right) \geq \beta_{\alpha,\delta}(n)/2\}.$$

Combining this with a union bound we obtain

$$\begin{aligned}
\mathbb{P}_{\boldsymbol{\theta}}^{\mathcal{A}_\delta} &\left(\exists t \in \mathbb{Z}_{>0} : N_a(t)D\left(\hat{\mu}_a(N_a(t)) \parallel \mu_a\right) \geq \beta_{\alpha,\delta}(t)/2\right) \\
&\leq \mathbb{P}_{\theta_a}\left(\exists t \in \mathbb{Z}_{>0} : tD\left(\hat{\mu}_a(t) \parallel \mu_a\right) \geq \beta_{\alpha,\delta}(t)/2\right) \\
&\leq \sum_{t=1}^{\infty} \mathbb{P}_{\theta_a}\left(D\left(\hat{\mu}_a(t) \parallel \mu_a\right) \geq \frac{\beta_{\alpha,\delta}(t)}{2t}\right).
\end{aligned}$$

We focus on upper bounding

$$\mathbb{P}_{\theta_a}\left(D\left(\hat{\mu}_a(t) \parallel \mu_a\right) \geq \frac{\beta_{\alpha,\delta}(t)}{2t},\ \hat{\mu}_a(t) \geq \mu_a\right).$$

Let $\mu_{a,t}$ be the unique (due to Corollary 1) solution (if no solution exists then the probability is already zero) of the equations

$$D\left(\mu_{a,t} \parallel \mu_a\right) = \frac{\beta_{\alpha,\delta}(t)}{2t}, \quad \text{and} \quad \mu_a \leq \mu_{a,t} \leq M.$$

Then the combination of Corollary 1 and Theorem 2 gives

$$\mathbb{P}_{\theta_a}\left(D\left(\hat{\mu}_a(t) \parallel \mu_a\right) \geq \frac{\beta_{\alpha,\delta}(t)}{2t},\ \hat{\mu}_a(t) \geq \mu_a\right) = \mathbb{P}_{\theta_a}\left(\hat{\mu}_a(t) \geq \mu_{a,t}\right) \leq \frac{\delta}{D}\frac{1}{t^\alpha}C^2.$$

We further upper bound the constant $c(P_{\mu_a})$ by $c(P)^2$ using Lemma 12, in order to obtain a uniform upper bound for any Markovian arm coming from the family.

A similar bound holds true for

$$\mathbb{P}_{\theta_a}\left(D\left(\hat{\mu}_a(t) \parallel \mu_a\right) \geq \frac{\beta_{\alpha,\delta}(t)}{2t},\ \hat{\mu}_a(t) \leq \mu_a\right).$$

The conclusion follows by summing up over all $t$ and using the simple integral based estimate

$$\sum_{t=1}^{\infty} \frac{1}{t^\alpha} \leq \frac{\alpha}{1-\alpha}.$$

$\square$

Embarking on the proof of the fact that the $(\alpha,\delta)$-Track-and-Stop strategy is $\delta$-PC we first show that the error probability is at most $\delta$ no matter the bandit model.

**Proposition 3.** *Let $\boldsymbol{\theta} \in \boldsymbol{\Theta}$, $\delta \in (0,1)$, and $\alpha > 1$. Let $\mathcal{A}_\delta$ be a sampling strategy that uses an arbitrary sampling rule, the $(\alpha,\delta)$-Chernoff's stopping rule and the best sample mean decision rule. Then,*

$$\mathbb{P}_{\boldsymbol{\theta}}^{\mathcal{A}_\delta}(\tau_{\alpha,\delta} < \infty, \hat{a}_{\tau_{\alpha,\delta}} \neq a^*(\boldsymbol{\mu})) \leq \delta.$$

*Proof.* The following lemma which is easy to check, and its proof is omitted, will be useful in our proof of Proposition 3.

**Lemma 13.** *The generalized Jensen-Shannon divergence*

$$I_a(\mu,\lambda) = aD\left(\mu \parallel a\mu + (1-a)\lambda\right) + (1-a)D\left(\lambda \parallel a\mu + (1-a)\lambda\right), \text{ for } a \in [0,1]$$

*satisfies the following variational characterization*

$$I_a(\mu,\lambda) = \inf_{\mu' < \lambda'} \{aD\left(\mu \parallel \mu'\right) + (1-a)D\left(\lambda \parallel \lambda'\right)\}.$$

If $\tau_{\alpha,\delta} < \infty$ and $\hat{a}_{\tau_{\alpha,\delta}} \neq a^*(\boldsymbol{\mu})$, then there $\exists t \in \mathbb{Z}_{>0}$ and there $\exists a \neq a^*(\boldsymbol{\mu})$ such that $Z_{a,a^*(\boldsymbol{\mu})}(t) > \beta_{\alpha,\delta}(t)$. In this case we also have

$$
\begin{aligned}
\beta_{\alpha,\delta}(t) &< Z_{a,a^*(\boldsymbol{\mu})}(t) \\
&= N_a(t) D\left(\hat{\mu}_a(N_a(t)) \,\|\, \hat{\mu}_{a,a^*(\boldsymbol{\mu})}(N_a(t), N_{a^*(\boldsymbol{\mu})}(t))\right) + \\
&\qquad N_{a^*(\boldsymbol{\mu})}(t) D\left(\hat{\mu}_{a^*(\boldsymbol{\mu})}(N_{a^*(\boldsymbol{\mu})}(t)) \,\|\, \hat{\mu}_{a,a^*(\boldsymbol{\mu})}(N_a(t), N_{a^*(\boldsymbol{\mu})}(t))\right) \\
&= (N_a(t) + N_{a^*(\boldsymbol{\mu})}(t)) I_{\frac{N_a(t)}{N_a(t) + N_{a^*(\boldsymbol{\mu})}(t)}}(\hat{\mu}_a(N_a(t)), \hat{\mu}_{a^*(\boldsymbol{\mu})}(N_{a^*(\boldsymbol{\mu})}(t))) \\
&= \inf_{\mu'_a < \mu''_a} \left\{ N_a(t) D\left(\hat{\mu}_a(N_a(t)) \,\|\, \mu'_a\right) + N_{a^*(\boldsymbol{\mu})}(t) D\left(\hat{\mu}_{a^*(\boldsymbol{\mu})}(N_{a^*(\boldsymbol{\mu})}(t)) \,\|\, \mu''_a\right) \right\} \\
&\leq N_a(t) D\left(\hat{\mu}_a(N_a(t)) \,\|\, \mu_a\right) + N_{a^*(\boldsymbol{\mu})}(t) D\left(\hat{\mu}_{a^*(\boldsymbol{\mu})}(N_{a^*(\boldsymbol{\mu})}(t)) \,\|\, \mu_{a^*(\boldsymbol{\mu})}\right),
\end{aligned}
$$

where the third equality follows from the variational formula for the generalized Jensen-Shannon divergence given in Lemma 13, and the last inequality follows from the fact that $\mu_a < \mu_{a^*(\boldsymbol{\mu})}$.

This in turn implies that, $\beta_{\alpha,\delta}(t)/2 < N_a(t) D\left(\hat{\mu}_a(N_a(t)) \,\|\, \mu_a\right)$, or $\beta_{\alpha,\delta}(t)/2 < N_{a^*(\boldsymbol{\mu})}(t) D\left(\hat{\mu}_{a^*(\boldsymbol{\mu})}(N_{a^*(\boldsymbol{\mu})}(t)) \,\|\, \mu_{a^*(\boldsymbol{\mu})}\right)$. Therefore by union bounding over the $K$ arms we obtain

$$
\begin{aligned}
\mathbb{P}_{\boldsymbol{\theta}}^{\mathcal{A}_\delta}&(\tau_\delta < \infty, \hat{a}_{\tau_\delta} \neq a^*(\boldsymbol{\mu})) \\
&\leq \sum_{a=1}^{K} \mathbb{P}_{\boldsymbol{\theta}}^{\mathcal{A}_\delta}\left(\exists t \in \mathbb{Z}_{>0} : N_a(t) D\left(\hat{\mu}_a(N_a(t)) \,\|\, \mu_a\right) \geq \beta_{\alpha,\delta}(t)/2\right).
\end{aligned}
$$

The conclusion now follows by applying Lemma 3. $\qquad\square$

*Proof of Proposition 1.*
Following the proof of Proposition 13 in Garivier and Kaufmann (2016), and observing that in their proof they show that $\tau_{\alpha,\delta}$ is essentially bounded we obtain that

$$
\mathbb{E}_{\boldsymbol{\theta}}^{\mathcal{A}_\delta}[\tau_{\alpha,\delta}] < \infty.
$$

This combined with Proposition 3 establishes that the $(\alpha,\delta)$-Track-and-Stop strategy is $\delta$-PC. $\qquad\square$

*Proof of Theorem 3.*
Finally for the proof the sample complexity of the $(\alpha,\delta)$-Track-and-Stop strategy in Theorem 3 we follow the proof of Theorem 14 in Garivier and Kaufmann (2016), where we substitute the usage of the law of large numbers with the law of large numbers for Markov chains, and in order to establish their Lemma 19 we use our concentration bound in Theorem 2. $\qquad\square$