[Reviews · NeurIPS 2019]

Reviewer 1



In general the paper is well written and easy to follow. I have couple questions about the results of the paper. See the comments below. The main concern is on the significance of the results. In particular, are the results in Section 2 about the reparametrization, and the result in Section 3 about the concentration of Markov chain mean estimation significant? The results look very interesting to me, but I am not familiar with this field. Other reviewers may have better judgement. Another concern is that the proposed algorithm needs the knowledge of P, as it is used to compute the stopping rule. This dependence makes the algorithm much less practical. Questions about the results: 1. Is the R_a term natural in Lemma 2? It seems to be a bit technical 2. Is the lower bound result in Section 3 under the same conditions with other references? For example, compared to [DZ98] and [Lez98]? 3. Does the algorithm need to know \phi? Minor comments: 1. Line 81-83: \mathcal{X} and \mathcal{S} are not unified. 2. Line 91. What is y_m? 3. Line 205, P_{(q,\mu)} is not defined. 4. Line 248 and 250, how are \hat{\mu}_a(N_a(t)) and \hat{\mu}_a(t) defined? ================ After rebuttal: Thanks for the reply. The rebuttal confirms that the algorithm needs some sort of knowledge of P. I think it needs some further justification why this is not a serious concern, or it should be stated explicitly. As it is also pointed out by Reviewer #4, the claims in section 3 are misleading. The comparison of The results of [Lez98], [DZ98], and Theorem 1 should be more rigorous and careful. I will keep my original score.

Reviewer 2



This work extends the problem of optimal best arm identification to the Markovian setting. Similarly to the iid case, each arm is still parametrized by a single parameter which can here be taken to be the stationary mean (exponential family of Markov chains), and a fair amount of structure is therefore assumed. For example the law of all arms is generated by the same Markov kernel (generator of the exponential family), and must be known for the algorithm to yield a priori guarantees and a valid stopping rule. The paper is of theoretical nature, is extremely well written and uses proper mathematical formalism. Its structure is doctrinal, allowing the reader to understand the underlying family of distributions, the bandit problem, main developed tools, and parse the results efficiently. Although the reviewer has no reason to doubt the results, a certain number of clarifications would be welcome here, especially with regards to Section 3. P(S) is defined at L.92 as the set of chains (identified with their Markov matrix) that satisfy a collection of irreducibility and other structural properties that are tied to a previously defined function φ. However, in the statement of Theorem 1, P(S) is introduced prior to φ. This begs the question of whether Theorem 1 is as general as currently stated. If indeed there is a strong connection between the function evaluated on the chain and the transition matrix for the theorem to hold, it would be uncalled for the author to claim general improvements over the references at L.163-164. In the proof of lemma 1, at L.373-374 the reader is referred to [Lax07], which guarantees existence of an eigenvector for the PF root that ’depends differentiably’ on the parameter, but also cautions the reader against the fact that not all such vectors might work. Is it therefore clear that the chosen u and v (where u sums to 1 and with v satisfying a prescribed inner product with u) are indeed sufficiently differentiable ? === Minor comments === L.91 ’y_m’ should be y L.135 ’the the’ L.166-167 The transpose notation is perhaps a bit too liberal here, as it refers to the adjoint of P in L2(π). L.371 ’eigenvectors’ L.375 Missing a constant term in log P(x,y) (that will indeed vanish upon taking the derivative) L.379 Last term should apply on x and not y. L.380 The infinitesimal term in theta is sub-scripted. L.387 Could you explain how dependence of this ratio in theta implies the ’Therefore’ ? L.461 ’eigenvalue’ L.488 Can you add intermediary steps for the second equality ? ==== AFTER REBUTTAL ==== The reviewer thanks the authors for their detailed answers. As argued by the authors, concentration inequalities are handy tools, but as such it is important for the reader to also quickly understand when they apply best, and when they don't. This coupling between φ and P could lead to a very interesting discussion and perhaps new insight. The reviewer keeps his score: an accept 7/10.

Reviewer 3



- It is not clear to me why one should care about a problem introduced by this paper. The authors put no words on introducing the physical background of the markov reward setting. I think motivation is important especially when the setting is not something that naturally exists: the scalar-exponential-family-parameterized markov chain looks artificial. It would help if the authors can explain why such setting is interesting to study. - It seems to me that given the concentration result in line 148, everything (lower bound, algorithm, and upper bound) directly follows [GK16]. It is not clear to me if there is anything interesting brought by the markov reward setting, regarding the problem structure revealed by the bound, challenges or techniques in the analysis. - The writing structure needs improvement. The actual problem setting is introduced on page 5 and even after reading the setting it takes me quite while to figure out what are the arms and what are the rewards. That said, the writing of theoretical statements looks clear. -------------------- update after reading the rebuttal: The authors pointed out a contribution I missed in my initial review - the finite sample concentration result. Although I still think the motivation for the problem setting is not very convincing and the results are a bit unsurprising from a practical point of view, I appreciate the technical clarity of this paper and that the authors work through all the theoretical results in a new best arm identification setting. I have increased my score accordingly.

[Author Response · NeurIPS 2019]

Reviewer # 6, the motivation is coming from the many significant applications of multi-armed bandit (MAB) models, e.g. clinical trials, ad placement, adaptive routing etc. For instance in a clinical trials setting it is natural to try to model the assertion: 'if a particular drug was successful the $n$-th time it's used, then it is very likely that it will be successful again the $n + 1$-th time it's used'. This assertion requires precisely the notion of Markovian dependence, and it can not be captured using the i.i.d. model. Through our work we increase the expressive power of MAB models and make it possible to express assertions like the aforementioned. Our work is technical and follows the typical approach of mathematical research where one generalizes results in broader mathematical situations. Also in the survey [BCB12] one can find references to works where the objective is regret minimization rather than best arm identification, involving rested and restless Markovian MAB models which have hundreds of citations. Moreover we disagree with the statement that the exponential family of stochastic matrices seems artificial. This is because many well established works in i.i.d. MAB, such as [GK16], as well as the work of Cappé et al on KL-UCB strategies, build on the notion of an exponential family of probability distributions. An exponential family of stochastic matrices is a generalization of an exponential family of finitely supported probability distributions, and so we essentially assume the same type of structure as many other major works that deal with the i.i.d. case.

Reviewer # 6, we strongly believe that the value of this work is coming precisely from its technical depth and the new tools introduced. The reason that Markovian bandits are much less studied compared to i.i.d. bandits is that the tools that probability theory has to offer in the case of Markovian dependence are so much less developed compared to the i.i.d. tools. We're the first to take the notion of an exponential family of stochastic matrices, initially introduced in the context of large deviations for Markov chains, and import it in the learning theory community. Moreover, in order to use it as the foundation of our Markovian MAB model we develop many technical properties of the family. One of the most important contributions of this work is the Chernoff type bound for Markov chains, we discuss it further in the next paragraph but here we note that reviewer # 6 incorrectly states that this is an asymptotic concentration result, while our result is non asymptotic. For the Markovian MAB lower bound our contribution is two fold. We first add the technical pieces needed in order to extend the standard change of measure argument to the Markovian setting, i.e. the regeneration argument for Markov chains. Moreover, we simplify the argument of [GK16] where they invoke some 'transportation Lemma', by observing that this so called 'transportation Lemma' is nothing more than a special case of the well known data processing inequality from information theory. The upper bound analysis indeed follows [GK16] quite closely, but even at this part we are able to fix a broken argument from [GK16]. In particular after contacting the second author of [GK16] we have verified that the last part of the proof of their Proposition 12 is wrong, and their $C$ has to depend on $\delta$ rendering their whole result shaky.

Reviewer # 3, for the Markovian bandit problem we take $S \subset \mathbb{R}$ and in L.177 we take $\phi(x) = x$, so the rewards that we observe are the states. Moreover the learning algorithm does not need to know the whole generator $P$, but only some scalar quantity related to it. For instance when $P$ is positive $\max_{x,y,z} \frac{P(x,z)}{P(y,z)}$ is all we need, while when the arms are i.i.d. we do not need to know something. In any case the generator $P$ is being tilted exponentially by $\theta_1, \ldots, \theta_K$ in order to produce the $K$ Markovian arms, and the algorithm is agnostic of all the tilts and of all the stationary means.

Reviewer # 4, analyticity of the PF eigenvalue, left and right eigenvectors is guaranteed because the eigenvalue is simple and we impose the two normalization constraints on the eigenvectors so that they are unique and the 'only determined up to a scalar factor' issue that [Lax07] mentions doesn't occur. Reviewer # 4, thank you for your detailed comments!

Regarding our Chernoff bound for Markov chains, roughly speaking large deviations dictate that the true rate of exponential decay is given by a relative entropy representing an information projection. This relative entropy already implicitly conveys information about the eigengap of $P$, the mixing time of the chain, it is $\phi$ specific etc. Any other bound that has a different rate of exponential decay is suboptimal and leads to a Pinsker type inequality like the one in L.171-172 (reviewer # 3, this inequality holds under the assumptions of Theorem 3.3 from [Lez98]). Therefore our bound has the optimal exponent, and the only way of potentially improving it is by introducing a better constant in the prefactor or by generalizing it. Reviewer # 4, L.163-164 are meant to say that when our assumptions are active and the assumptions of some other work are active, then our bound is better. For instance, we can consider the important case of a strictly positive $P$ (no matter $\phi$). Then Theorem 3.3 from [Lez98] and our Theorem 1 are both in action, with ours dominating. Reviewer # 4, your observation is correct, in Theorem 1 we should have first fixed $\phi$ and then fix $P$, but this does not alter the optimality statement that we made above. To the best of our knowledge (following citations and consulting local experts) this constitutes a novel and optimal non asymptotic concentration result for Markov chains. Bounds of this type are of fundamental importance, not only in learning theory, e.g. MCMC, Markov decision processes etc, but also in general in the world of applied probability. For instance the work of [Gil93], which presents a suboptimal bound for reversible Markov chains accompanied with computer science applications, is coming from FOCS, (best TCS conference), and bounds inspired by the work of [Gil93] based on matrix perturbation theory without accompanying applications [Din95, Lez98, LP04], are coming from the Annals of Applied Probability (one of the best probability theory journals). In this work we're not only developing an optimal Chernoff bound for Markov chains using new techniques, but we also apply this in order to resolve the complexity of the best Markovian arm problem.

[Meta-Review · NeurIPS 2019]

The paper extends the i.i.d. best arm identification problem to a special Markovian setting. The technical tool is a new Chernoff-type inequality for a special class of Markov chains, which allows transferring the method for the iid problem to the considered Markov case. While the reviewers found the results interesting, they raised a number of questions, which should be addressed in the final version. These, include, among others, requiring a better motivation (note that the motivation given in the response does not apply, since there the chain is not ergodic), including the limitations of the reward models, an argument why some it is not a problem that some extra information about the reward model is needed for the algorithm, etc.